# Cortex-wide transcranial localization microscopy with fluorescently labeled red blood cells

Quanyu Zhou [1,2,5], Chaim Glück [1,3,5], Lin Tang[1,2], Lukas Glandorf [1,2], Jeanne Droux[3,4], Mohamad El Amki [3,4], Susanne Wegener[3,4], Bruno Weber [1,3], Daniel Razansky [1,2,3] ✉ & Zhenyue Chen [1,2] ✉

Large-scale imaging of brain activity with high spatio-temporal resolution is crucial for advancing our understanding of brain function. The existing neuroimaging techniques are largely limited by restricted field of view, slow imaging speed, or otherwise do not have the adequate spatial resolution to capture brain activities on a capillary and cellular level. To address these limitations, we introduce fluorescence localization microscopy aided with sparsely-labeled red blood cells for cortex-wide morphological and functional cerebral angiography with 4.9 μm spatial resolution and 1 s temporal resolution. When combined with fluorescence calcium imaging, the proposed method enables extended recordings of stimulus-evoked neuro-vascular changes in the murine brain while providing simultaneous multiparametric readings of intracellular neuronal activity, blood flow velocity/direction/volume, and vessel diameter. Owing to its simplicity and versatility, the proposed approach will become an invaluable tool for deciphering the regulation of cortical microcirculation and neurovascular coupling in health and disease.

Simultaneous neuro-vascular imaging with high spatio-temporal resolution on a large scale is crucial for advancing our understanding of brain function and disease progression. Task-evoked neuron activation causes the initial increase of local metabolic consumption, further triggering a down-stream regulation of cerebral blood flow to ensure an excess supply of oxygen and glucose to the active brain regions. This phenomenon, known as neurovascular coupling, illustrates the intricate interplay between neural activity and the accompanying hemodynamic changes[1]. Owing to the lack of suitable imaging tools, the spatial and temporal features of this coupling at the whole cortex scale are not yet fully understood. Likewise, the relationship between increased perfusion and alterations in blood velocity and vessel diameter across different types of vessels remains largely unexplored.

Existing neuroimaging tools map brain activation either through direct measurement of neural activity or indirect assessment of hemodynamics. Microscopic neuroimaging techniques, as exemplified by two-photon microscopy (2PM) in conjunction with genetically encoded calcium indicators (GECIs), serve as a standard approach for directly observing neural activity at a cellular resolution[2]. However, the typical scanning rate of 2PM poses challenges in capturing blood flow across the entire field of view (FOV) simultaneously. As an alternative, one-dimensional (1D) line-scanning along the centerline of selected vessel combined with kymograph-based analysis is employed to provide the flow velocity[3]. The line-scanning rate ultimately constrains the maximum detectable flow velocity, which is typically below 10 mm/s. Furthermore, the limited FOV (typically < 1 mm) of conventional 2PM

[1]Institute of Pharmacology and Toxicology, Faculty of Medicine, University of Zurich, Zurich, Switzerland. [2]Institute for Biomedical Engineering, Department of Information Technology and Electrical Engineering, ETH Zurich, Zurich, Switzerland. [3]Zurich Neuroscience Center, Zurich, Switzerland. [4]Department of Neurology, University Hospital and University of Zurich, Zurich, Switzerland. [5]These authors contributed equally: Quanyu Zhou, Chaim Glück. ✉e-mail: daniel.razansky@uzh.ch; zhenyue.chen@uzh.ch

further restricts its applicability for imaging the neural- and vascular-network as a whole.

Most meso- and macro-scopic neuroimaging tools infer brain activities from indirect measurement of hemodynamic response. However, the mechanisms and parameters provided by different modalities vary significantly. Owing to its noninvasiveness and brain-wide access volume, functional magnetic resonance imaging (fMRI) has been extensively used to study brain function and connectivity by measuring blood oxygen level dependent (BOLD) signals[4,5]. However, the complex origin of BOLD signals makes it challenging to unequivocally decouple the blood flow parameters (cerebrovascular blood volume, CBV; cerebrovascular blood flow, CBF) from oxygenation-related (oxyhemoglobin, HbO; deoxyhemoglobin, HbR) readings[6]. In contrast, functional ultrasound imaging (fUS) is chiefly sensitive to CBV changes reflected in the measured power doppler signals[7–9]. Similarly, optical coherence tomography (OCT) can measure CBF across multiple vessels by leveraging Doppler effect through repetitive B-scans, which renders velocity maps in either coronal or sagittal views. However, functional neuroimaging with OCT across the extensive vascular network covering the entire mouse cortex remains an unmet need due to its limited scanning speed[10]. Laser speckle contrast imaging (LSCI), a label-free modality for CBF mapping, can achieve centimeter-scale FOV in the transverse plane with sufficient temporal resolution for observing hemodynamic changes. However, LSCI suffers from limited spatial resolution with blood flow quantification remaining out of reach with this method due to the equivocal model of speckle contrast[11]. Both functional optoacoustic tomography (OAT)[12,13] and intrinsic signal optical imaging (ISOI)[14] can provide blood oxygenation readings based on optical absorption coefficient difference of HbO and HbR when employing spectral unmixing methods. Simultaneous neuro-vascular imaging was further enabled by combining these methods with epifluorescence calcium imaging[14–16]. Overall, these approaches suffer from inadequate spatio-temporal resolution, impeding the quantitative measurement of hemodynamic responses, particularly when it comes to high-order vessel branches such as arterioles and venules. Functional ultrasound localization microscopy (fULM) allows for transcranial imaging of cerebrovascular flow with enhanced resolution by tracking intravenously injected microbubbles[17], thus enabling the observation of stimulus-evoked hemodynamic responses deep in mammalian brain. However, real-time observations with fULM are commonly limited to single cross-sections in the sagittal or coronal planes whilst direct measurement of neural activity remains inaccessible with this modality.

Here, we propose a widefield fluorescence localization microscopy (WFLM) method aided with sparsely-labeled red blood cells (RBCs) to achieve cortex-wide morphological and functional vascular imaging in the murine brain with 4.9 μm spatial resolution and 1 s temporal resolution. The RBC-aided WFLM technique is based on continuous localization and tracking of intravenously injected fluorescent RBCs. We applied the proposed method to monitor hemodynamic changes in the murine brain during peripheral sensory stimulation by providing multiparametric readings including vessel diameter and blood flow velocity/direction/volume across different types of vessels. Furthermore, by integrating the proposed method with epifluorescence calcium imaging, we demonstrate simultaneous measurement of hemodynamic and calcium responses during hindpaw and whisker stimulations thus depicting various stimulus-evoked neuro-vascular activation patterns. We further explore the potential of our proposed method in transcranial neuro-vascular imaging to correlate the spatial extent and activation intensity of hemodynamic responses in both contralateral and ipsi-lateral hemispheres. This approach is expected to bridge the gap between mesoscopic and microscopic neuroimaging methods, shedding light on the interplay between neural activity and hemodynamic responses.

## Results

### Methodology of RBC-aided WFLM and characterization

The basic system configuration is illustrated in Fig. 1a. It follows a typical widefield fluorescence microscopy design[18], incorporating a 660 nm excitation source and a high-speed camera capable of achieving frame rates of up to 833 Hz (see online Methods for details). An additional fluorescence channel, featured with a 488 nm excitation source and an extra detection camera serves as an add-on module for GCaMP imaging. RBC labeling was achieved using a lipophilic DiD fluorescent dye to stain the cell membrane ex vivo (Fig. 1a, right), as validated with a commercial confocal microscope (Fig. 1b). To access the imaging performance of the proposed system, DiD-stained RBCs were intravenously injected into a Thy1-GCaMP6f mouse after cra-niotomy, followed by 6 min widefield image recording at a frame rate of 400 Hz. The sparsity of stained RBCs in the total blood can be controlled by adjusting the number of stained RBCs for injection, while the localization and tracking accuracy is influenced by magnification and frame rate of the imaging system. A high-resolution intensity map of murine cerebral vasculature was rendered through continuous localization and tracking of circulating RBCs (Fig. 1c and Supplementary Fig. 1a–d). As a byproduct of the tracking process, color-encoded flow velocity and direction maps were also obtained (Fig. 1d, e). The flow direction map facilitates the differentiation between veins and arteries (Fig. 1e and Supplementary Fig. 1e, f). The superior spatial resolution is then corroborated by the zoom-in view (Fig. 1f). Two neighboring vessels with an interval of 4.9 μm could be distinguished unambiguously, representing the practical spatial resolution of the system in vivo (Fig. 1f, right). Note that theoretical resolution of localization-based imaging methods primarily hinges on the signal-to-noise ratio (SNR) and spatial sampling rate of the system[19]. Practically, the localization image quality is also related to the quantity of tracked stained RBCs (Supplementary Fig. 2). We further validated the imaging depth by imaging the same mouse brain with RBC-aided WFLM and 2PM (Supplementary Fig. 3). Through direct comparison of the same vessels imaged by both methods, the deepest vessel detected by RBC-aided WFLM is located ~200 μm below the brain surface.

Taking advantage of the original frame rate of 400 Hz, the temporal velocity profile along the center lines of representative artery, vein, and capillary branches was tracked during reconstruction (Fig. 1g). The relatively high RBC flux in the artery and vein allows for an effective (compounded) frame rate of 20 Hz after localization reconstruction whereas the capillary was reconstructed at a temporal resolution of 1 s due to relatively low RBC flux (Fig. 1g). For the selected artery and vein, the 20 Hz effective frame rate facilitates the observation of pulsatile flow after fitting with a sinusoidal function. Fourier transform revealed dominant frequency components at 4.63 Hz and 4.61 Hz in the artery and vein, respectively (Fig. 1h), which closely align with the real-time measured heart rate of 277 beats per minute (bpm). We extended the velocity pulsatility analysis to $n = 50$ vessel branches with diameters ranging from 12.6 to 119.0 μm (including $n = 29$ arteries and $n = 21$ veins) with the calculated dominant frequency being $4.61 \pm 0.03$ Hz (Fig. 1i). A weak correlation between the dominant frequency and the vessel diameter was corroborated with Pearson's correlation coefficient of 0.11, underscoring the capability of RBC-aided WFLM to accurately quantify the flow dynamics across different vessel diameters.

### Capillary-level multiparametric mapping of sensory stimulation-evoked hemodynamic responses

We further employed RBC-aided WFLM to map the hemodynamic changes in the mouse brain during sensory stimulation. To check hemodynamic changes across different vessel types (i.e., artery, vein, and capillary), localization compound images were reconstructed at an effective frame rate of 1 Hz. An electrical stimulation paradigm including 12 repeated cycles, adapted from previous reports[16] was applied to the

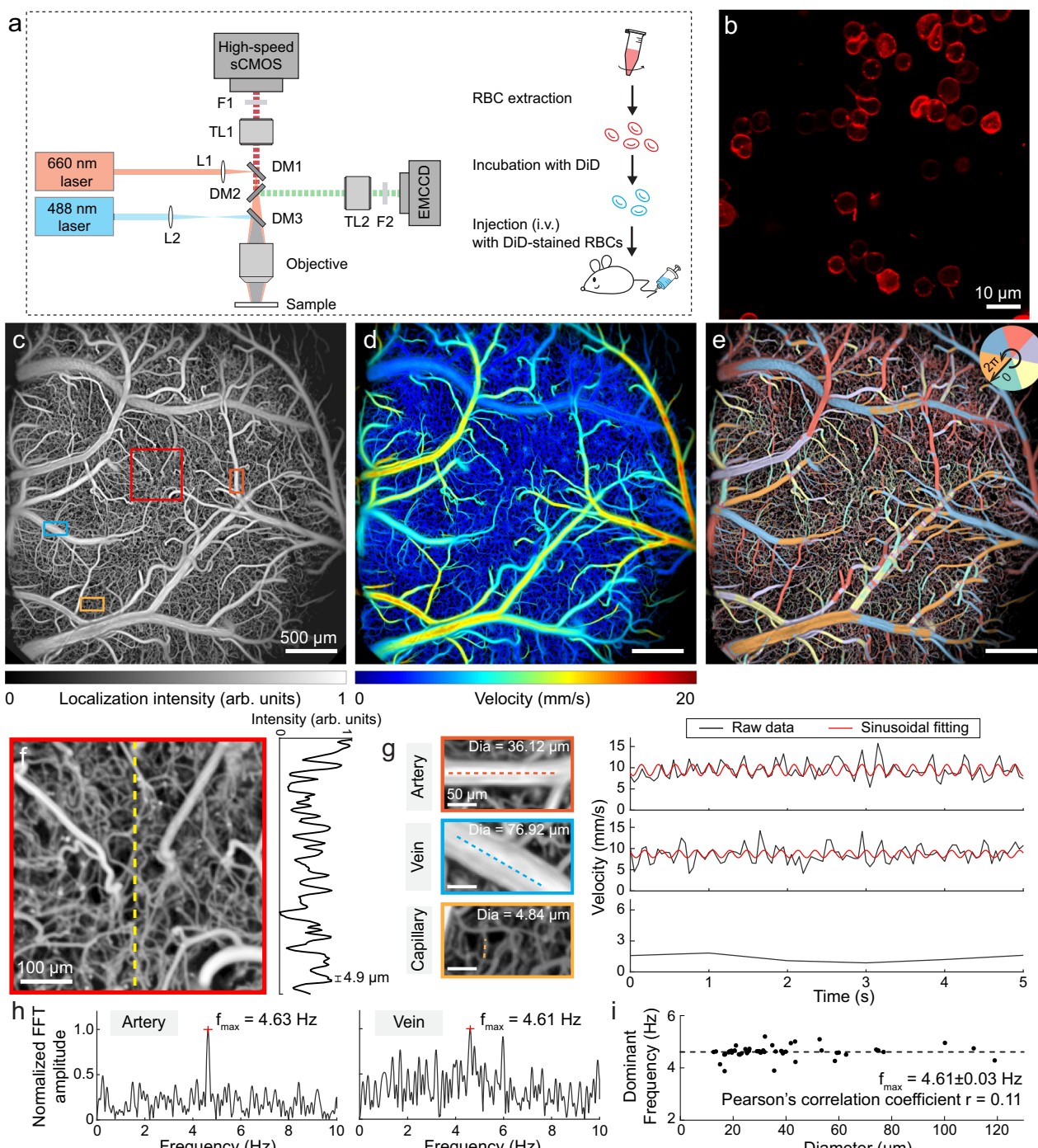

**Fig. 1 | System design and imaging performance of RBC-aided WFLM.**
**a** Experimental setup for RBC-aided WFLM imaging based on a widefield excitation/detection of sparsely-labeled RBCs circulating in the mouse brain. An optional 488 nm excitation channel is available for GCaMP detection. **b** Confocal image of DiD-stained RBCs. The experiment was repeated independently for three times with similar results. **c** Localization reconstructed structural map of the mouse brain. **d, e** Color-encoded flow velocity and direction maps, respectively. **f** Zoom-in view of the ROI indicated by the red square in **c**, with line profile along the yellow dashed line shown. **g** Average velocity temporal profiles from center lines situated on a representative artery, vein, and capillary (marked with rectangles in **c**). Vessel diameters of selected branches were labeled. **h** Fourier transform analysis of velocity temporal profiles in artery and vein. **i** Relationship between the calculated dominant frequency and vessel diameter. L1-2, optical lenses; DM1-3, dichroic mirrors; TL1-2, tube lenses; F1-2, filters; Dia, diameter; FFT, fast Fourier transform. Representative data from one mouse are shown. Source data of Fig. 1f–i are provided as a Source Data file.

right hindpaw of an anesthetized C57BL/6 mouse with cranial window. The activation region of the mouse was firstly mapped with multispectral ISOI. Capitalizing on the superior spatio-temporal resolution, a super-resolved localization intensity map could then be rendered post intravenous injection of DiD-stained RBCs (Fig. 2a). Through pixelwise calculations of Pearson's correlation coefficient between the velocity time course and an averaged stimulation induced response curve, a correlation map of flow velocity was obtained, revealing a spatial activation pattern (Fig. 2b) and activation intensity (Fig. 2c) during functional hyperemia. Additionally, the temporal activation pattern characterized by pixelwise time-to-peak (TTP) was obtained by calculating the latency between the starting time of the stimulation and the

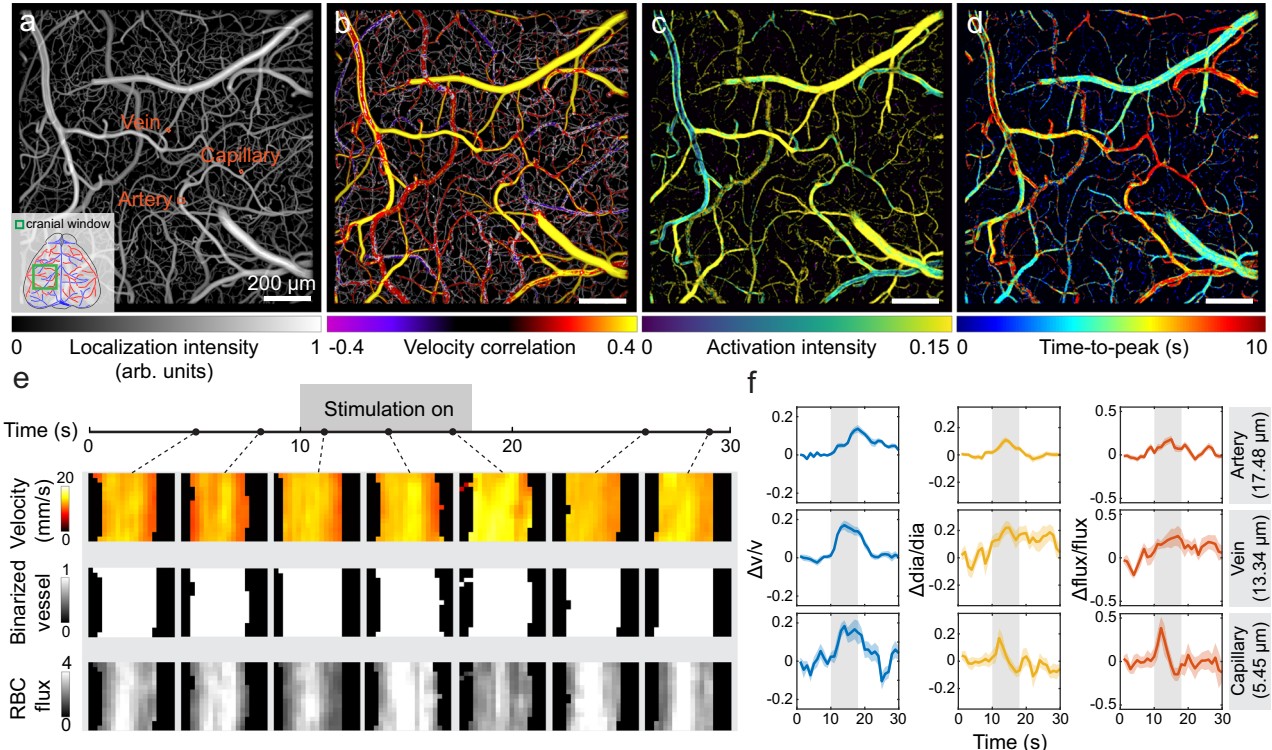

**Fig. 2 | Multiparametric mapping of hindpaw stimulation-evoked functional hyperemia in the murine brain using RBC-aided WFLM. a** Localization structural map of cerebral vasculature in an anesthetized C57BL/6 mouse under hindpaw stimulation. **b** Flow velocity correlation map obtained by calculating the Pearson's correlation coefficient between the velocity temporal profile and local averaged hemodynamic response curve. **c** Activation intensity map of velocity response. **d** TTP map depicting the onset timing of velocity activation. **e** Time-lapse velocity map, binarized structural map, and RBC flux map of the artery labeled in **a** during a representative stimulation cycle. **f** Temporal profiles of velocity, diameter, and RBC flux change from a representative artery, vein, and capillary marked in **a**. Vessel diameters of selected branches were labeled. Data are presented as the mean ± s.e.m. Representative data from one mouse are shown. Source data of Fig. 2f are provided as a Source Data file.

velocity peak in each vessel (Fig. 2d). Figure 2e depicts time-lapse images of velocity map, binarized vessel structural map, and RBC flux map from a selected region of interest (ROI) situated in an artery during a representative stimulation cycle, providing both structural and functional information on hemodynamic changes. Averaged blood flow velocity, vessel diameter, and RBC flux curves across all the stimulation cycles were further depicted from a representative artery, vein, and capillary (Fig. 2f and Supplementary Movie 1), revealing distinctive activation pattern in the vascular network. Statistical analysis revealed a significant increase in flow velocity, vessel diameter, and RBC flux upon stimulation in the selected vessels (Supplementary Fig. 4a, b). We further selected $n = 20$ arteries, $n = 10$ veins, and $n = 5$ capillaries for detailed analysis. In comparison, capillaries exhibit higher fractional changes in velocity and RBC flux than veins and arteries post hindpaw stimulation (Supplementary Fig. 4c), which is aligned with previous study[20]. Both arteries and capillaries show synchronized diameter increase (vasodilation) during stimulation while it is not as obvious in veins, potentially related to the activation of enwrapping smooth muscle cells and pericytes[21,22]. Interestingly, we observed that the artery diameter increase peaked significantly earlier ($3.95 \pm 0.28$ s) than the velocity increase ($7.20 \pm 0.57$ s), as depicted in Fig. 2f and Supplementary Fig. 4d, e, indicating that vasodilation in arteries plays a more important role in regulating the blood flow volume upon external stimulation. In contrast, veins and capillaries are featured with close TTP values for both velocity and diameter responses (Supplementary Fig. 4e).

**Visualizing the intricate interplay between neuronal activity and accompanying hemodynamic responses**
The widefield detection scheme of RBC-aided WFLM facilitates its combination with epifluorescence calcium imaging. We implemented a

two-channel fluorescence excitation/emission system (Fig. 1a) for concurrent neuro-vascular imaging to study neural and hemodynamic activation patterns under hindpaw and whisker stimulations. Right hindpaw stimulation was applied to anesthetized Thy1-GCaMP6f mice ($n = 5$ mice) after craniotomy. The GCaMP activation map, computed from a general linear model (GLM), was superimposed onto the original structural map in the GCaMP channel (Fig. 3a). GCaMP activation map was thresholded at 20% quantile to demarcate the activation region, a criterion maintained across all GCaMP datasets. Additionally, a high-resolution image was obtained through localization-based reconstruction of the image stack recorded in the DiD channel (Fig. 3b). Correlation map of flow velocity was calculated to quantify the hemodynamic activation pattern (Fig. 3c). Co-localized activation regions were observed in both the GCaMP activation map and velocity correlation map (Fig. 3a, c and Supplementary Fig. 5a). Figure 3d depicts the average temporal profile of GCaMP signals from two ROIs with one inside and one outside the GCaMP activation region, whereas only the former shows synchronized increase during stimulation. To assess functional hyperemia, vessel segments inside and outside the GCaMP activation region were selected. Importantly, arteries and veins are differentiated based on their different flow directions (Supplementary Fig. 5b, c). Vessel segments inside the GCaMP activation region showed a blood flow increase with significantly higher amplitudes ($7.5 \pm 1.0\%$), compared to that outside the region ($3.5 \pm 0.6\%$), as shown in Fig. 3e and Supplementary Fig. 5d, e. We further investigated the activation intensity and onset timing of hemodynamic response along the vessel tree within the GCaMP activation region by selecting six vessel branches on the same vein (Fig. 3f). As the order of vessel branch increases, the activation intensity increases while TTP value decreases (Fig. 3g and Supplementary Fig. 5f), suggesting a passive blood regulation originating from the

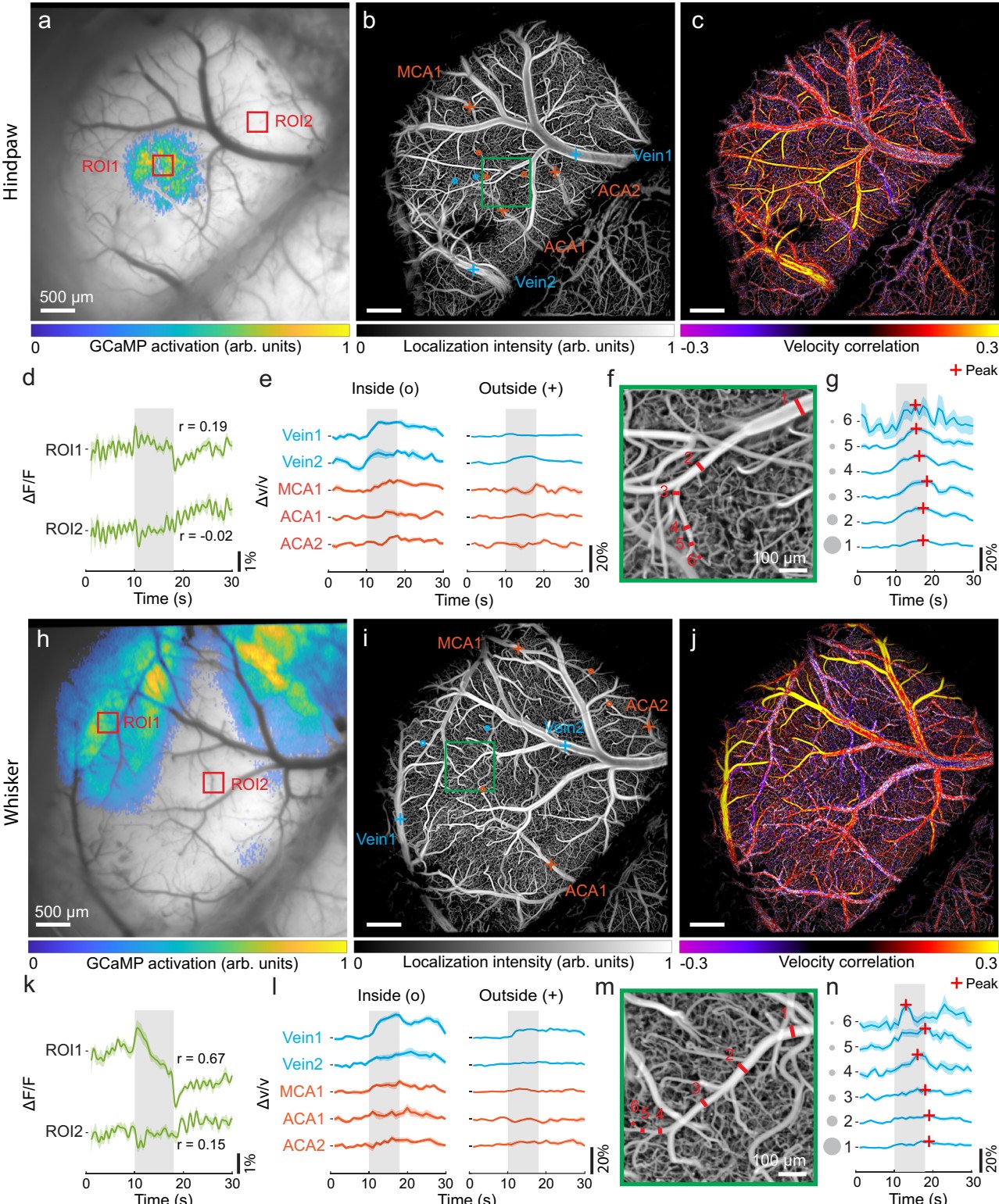

**Fig. 3 | Simultaneous neuro-vascular activation mapping in mice under hind-paw and whisker stimulations utilizing RBC-aided WFLM combined with epi-fluorescence calcium imaging. a–c** GCaMP activation map, localization intensity map, and velocity correlation map obtained from a Thy1-GCaMP6f mouse under right hindpaw stimulation. **d** Averaged GCaMP time courses within two ROIs inside and outside the GCaMP activation region, as indicated with red squares in **a**. **e** Averaged velocity time courses in five vessel segment pairs located inside and outside the GCaMP activation region selected from typical pial vessels, as marked in **b**. **f** Expanded view of the ROI labeled with green box in **b**. **g** Averaged velocity time courses in different segments along the same vein labeled in **f**. **h–j** GCaMP activation map, localization intensity map, and velocity correlation map collected from

another mouse under right whisker stimulation. **k** Averaged GCaMP time courses within two ROIs labeled in **h**. **l** Averaged velocity time courses in five vessel segment pairs located inside and outside the GCaMP activation region selected from typical pial vessels, as marked in **i**. **m** Zoom-in view of the ROI labeled with green box in **i**. **n** Averaged velocity time courses in different vessel segments in **m**. MCA, middle cerebral artery; ACA, anterior cerebral artery. Data are presented as mean ± s.e.m. Pearson's correlation coefficients were calculated between fluorescence signal time courses and the regressor built based on the GCaMP response function in **d** and **k**. The experiment was repeated independently in five mice with similar results. Source data of Fig. 3d, e, g, k, l, n are provided as a Source Data file.

higher-order branches. A similar phenomenon that the activation intensity increases with the branch order was also observed along an arterial tree (Supplementary Fig. 5g, h).

Following a similar procedure, Thy1-GCaMP6f mice (*n* = 5 mice) were imaged after stimulation of the right whisker. The GCaMP activation map (Fig. 3h) exhibits a different spatial distribution surrounding the middle cerebral artery (MCA), consistent with the mapping result using ISOI[23]. Localization intensity and velocity correlation maps (Fig. 3i, j) were rendered in the same way as shown previously with RBC-aided WFLM. As expected, the velocity correlation map displays a similar activation pattern to the GCaMP activation map (Fig. 3h, j, Supplementary Fig. 6a, and Supplementary Movie 2). GCaMP signal time course shows clear stimulus-evoked response within the activated region as compared to lack of response in the surrounding tissues (Fig. 3k). Similar to hindpaw stimulation, we observed hemodynamic changes in all selected vessel branches inside the activated region (9.7 ± 2.0%) and weak responses in the vessel branches outside this region (3.2 ± 1.2%), as depicted in Fig. 3l and Supplementary Fig. 6b–e. A decrease in activation intensities and an increase in TTP values were observed when moving from the high-order to low-order segment of a representative vein (Fig. 3m, n and Supplementary Fig. 6f). Changes in different vessel segments along the arterial tree is depicted in Supplementary Fig. 6g, h.

### Transcranial imaging of cortex-wide neuro-vascular activation

Next, we exploited the potential of RBC-aided WFLM for imaging cortex-wide neurovascular coupling transcranially (*n* = 4 mice). System's magnification ratio was decreased to 0.86 with a different combination of objective and camera lens to cover the entire cortex. The ability to access both hemispheres allows for the observation of differing responses from contralateral and ipsilateral sides of the brain. An anesthetized Thy1-GCaMP6f mouse with intact skull was imaged under sequential right and left hindpaw stimulations. Accordingly, the contralateral (opposite to the stimulation side) and ipsilateral sides in the brain were alternated. Following the localization-based reconstruction, a structural map with a mixture of skull and brain vessels was rendered (Fig. 4a). Predominant calcium response was observed on the contralateral side in the GCaMP activation maps (Fig. 4b, c). In the corresponding velocity correlation maps, hemodynamic activation was observed on both sides, with the contralateral side displaying a higher correlation coefficient (Fig. 4d, e).

We further analyzed the differential responses of the same region/ vessel when switching the stimulation from the right to left hindpaw. Temporal profiles of GCaMP signals from two ROIs situated on both hemispheres (indicated in Fig. 4b, c) reveal a more pronounced signal increase when the stimulation was applied on the opposite side of the limb, known as the brain lateralization (Fig. 4f). A similar phenomenon was observed in velocity time courses from five out of six representative vessels (Fig. 4f, bottom row and Supplementary Fig. 7). Note that in transcranial imaging, the presence of the skull introduces additional photon scattering which compromises the imaging SNR and thus localization accuracy. The diminished SNR complicates detection of sufficient stained RBCs to accurately depict hemodynamic changes in the capillaries. Nonetheless, our method retains the capability to detect activations in arteries and veins with diameters as small as ~15 μm (Supplementary Fig. 8). Statistical analysis reveals that veins displayed a significantly higher activation intensity from the contralateral side than the ipsilateral side (7.9 ± 1.6% vs. 3.8 ± 0.6%, *n* = 8 vessels) while arteries exhibited comparable responses under two conditions (4.8 ± 0.5% vs. 4.1 ± 0.6%, *n* = 11 vessels), as illustrated in Fig. 4g. Additionally, we calculated the velocity correlation coefficients of selected veins and arteries, where the correlation coefficients from the contralateral side were obviously higher than the ipsilateral side (Fig. 4h).

## Discussion

In this work, we introduce the RBC-aided WFLM approach for large-scale microcirculation mapping, offering simultaneous morphological and functional mapping of rodent brain activity. Its superior spatio-temporal resolution makes it an ideal neuroimaging tool for detecting stimulation-evoked hemodynamic changes, while providing multi-parametric readings down to the capillary level. The method offers unique capability to record distinct hemodynamic response profiles across vessel compartments and evaluate the precise onset of different events regulating blood flow. Concurrent neuro-vascular imaging was achieved when combining WFLM with epifluorescence calcium imaging. Co-localized neural and hemodynamic activation regions were observed for both whisker and hindpaw stimulations, exhibiting different activation patterns. The availability of GCaMP activation map guides the comparative analysis of hemodynamic responses in connected vessel branches located both inside and outside the GCaMP activation region. Furthermore, we demonstrate the potential of RBC-aided WFLM in transcranial neuro-vascular imaging across the entire cortex, facilitating the observation of varying hemodynamic responses from the same vessel under bilateral stimulation.

Compared to previous work[18,24] employing micrometer-sized fluorescent beads as sparse fluorescence emitters, the synthesized fluorescently-labeled RBCs exhibit significantly longer circulation time of up to tens of minutes. Moreover, blood flow can be assessed across the whole pial vasculature from arteries down to capillaries, ensuring robust functional readings of stimulus evoked brain activation. Their inherent biocompatibility, prolonged circulation lifespan, and mechanical properties akin to innate blood cells significantly augment the imaging performance in terms of reliability of flow velocity and direction estimations and spatio-temporal resolution. Furthermore, the deep-red fluorescence emission of DiD allows for deeper penetration. In future work, we envision incorporating alternative labeling dyes or antibodies with emission bands extending into the second near-infrared window[25–27] to attain an extended penetration range.

Recent developments in single- and multi-photon fluorescence microscopy techniques have improved spatial sampling rate through the implementation of temporal multiplexing[28], multifocal illumination[24,29–31], polygon scanner[32], thus potentially elevating the maximum detectable blood velocity limits. Yet, scanning microscopy techniques remain constrained by either a restricted FOV or lack of capability to offer quantitative functional readings. By utilizing a redesigned microscopic objective featuring a spherical intermediate plane and parallel acquisition with 35 sCMOS cameras, a real-time, ultra-large-scale, high-resolution (RUSH) method has been reported[33]. This approach is designed to overcome the limited space-bandwidth product, achieving ~1.2 μm resolution across centimeter-scale FOV. The video-rate frame rate further enables cortex-wide tracking of immune cells and vascular dynamic imaging in mice. However, highly invasive procedures such as craniotomy are inevitable in most cases to reach the brain. In contrast, RBC-aided WFLM is a highly parallel widefield modality with a simple and cost-effective optical design exploiting spatial sparsity introduced by labeled RBCs, making it particularly suitable for high-resolution large-scale functional neuroimaging in both cranial-windowed and transcranial scenarios, despite a diminished SNR caused by the highly heterogenous skull. In addition, random scattering events may lead to distortions in the point spread function (PSF), thus degrading localization accuracy with Gaussian fitting-based localization methods. Incorporating adaptive optics[34] could shape the wavefront of fluorescence emission from each single labeled RBC, thereby improving the localization accuracy and spatial resolution. Another potential strategy to enhance the imaging resolution and penetration depth in RBC-aided WFLM is to incorporate skull optical clearing methods, which has been demonstrated to improve imaging performance in multi-photon imaging[35,36].

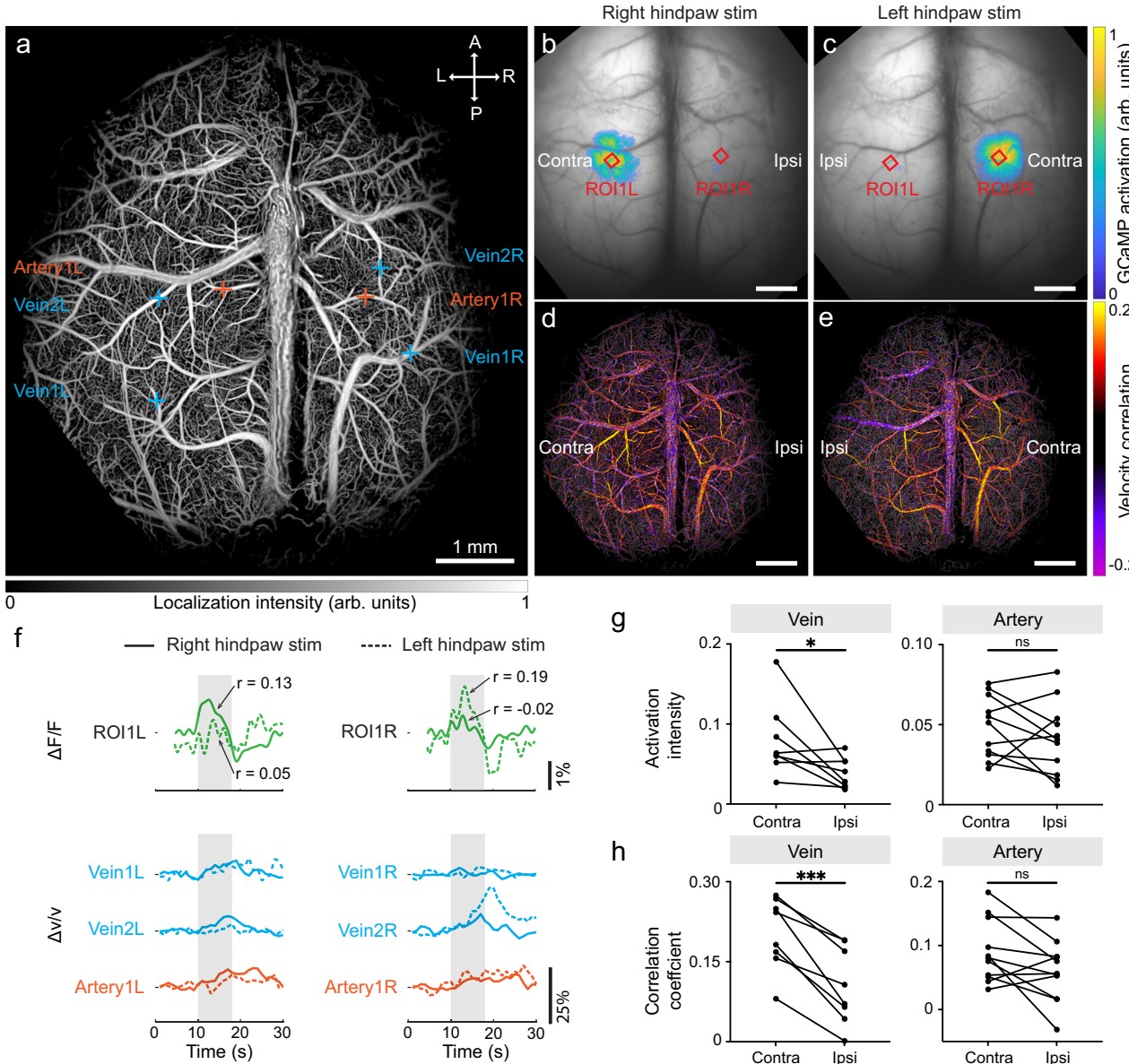

**Fig. 4 | Transcranial, cortex-wide neuro-vascular activation mapping in mice under hindpaw stimulations. a** Localization intensity map of a Thy1-GCaMP6f mouse during hindpaw stimulation. **b, c** GCaMP activation maps derived from right and left hindpaw stimulations, respectively. **d, e** Flow velocity correlation maps calculated from right and left hindpaw stimulations, respectively. **f** Temporal profiles of the GCaMP signal and flow velocity change within selected ROIs (indicated with red squares in **b, c**) or vessels (indicated with crosses in **a**). Pearson's correlation coefficients were calculated between fluorescence signal time courses and the regressor built based on the GCaMP response function. **g** Comparative

analysis of activation intensity of the same veins and arteries under right and left hindpaw stimulations. The calculated *P* values for vein and artery are 0.04 and 0.27, respectively. **h** Comparative analysis of correlation coefficient of the same veins and arteries under right and left hindpaw stimulations. The calculated *P* values for vein and artery are 0.0004 and 0.0504, respectively. Two-tailed paired *t*-test was used in **g** and **h** (ns, not significant, \**P* < 0.05, \*\**P* < 0.01, \*\*\**P* < 0.001). The experiment was repeated independently in four mice with similar results. Source data of Fig. 4f–h are provided as a Source Data file.

fULM techniques can image deeper brain regions[17]. However, a continuous injection of microbubbles is needed during the imaging session due to their fast clearance rate. Furthermore, due to its B-scan acquisition topology, fULM is limited to recording a single coronal slice at a time, thus hindering real-time visualization across horizontal brain planes. In contrast, the long circulation lifespan of RBCs does not require multiple contrast agent injections. More importantly, the cortex-wide transverse views provided by RBC-aided WFLM facilitates neurovascular studies into pial vascular network, thus complementing the fULM method. Recently, functional OAT[16,37,38] and optoacoustic microscopy (OAM)[39] techniques have been employed to monitor rodent brain activation by providing multiparametric hemodynamic

readings along with oxygen saturation. However, the effective spatio-temporal resolution of OAT scales with the imaged FOV, making it incapable to differentiate both structural and functional changes at capillary level. In principle, OAM can provide flow velocity maps on a single vessel level based on flow-induced decorrelation[40], Doppler effect[41], and Grüneisen relaxation effect[42]. Additional blood oxygenation readings could be achieved through the implementation of multiple-wavelength excitation[43] or pulse-width-based method[44]. However, the maximal detectable flow speed of OAM is limited to ~11 mm/s due to the restricted scan rate and lack of capacity to discriminate between the individual RBCs[45]. In addition, the contact-based nature of ultrasound detection complicates the design of

multimodal systems to combine OAM with fluorescence-based calcium imaging methods.

A number of technical improvements are envisioned for enhancing the proposed method's performance. A multi-view detection strategy[46] can be applied to add depth-resolved (3D) information and access penetrating vessels, which would allow for a more comprehensive analysis of hemodynamic responses in both lateral and axial directions. Another direction is to implement multispectral WFLM by utilizing various fluorescent dyes to label different blood cells, thus enabling highly multiplexed investigations on the interplay and flow dynamics. In addition, owing to its simplicity and compatibility, the method can be integrated with other neuroimaging techniques such as fMRI and functional OAT. While RBC-aided WFLM can provide information on both CBF and CBV with high spatio-temporal resolution, fMRI and OAT can provide additional information on multiple hemodynamic components, oxygen consumption, and anatomical information at the whole-brain level. Multimodal neuroimaging could then be exploited to decouple the influence of blood flow parameters on the oxygenation-related readings from fMRI and OAT, thereby delineating a comprehensive picture of neurovascular coupling.

In conclusion, we proposed an optical localization microscopy technique aided with fluorescently-labeled RBCs and demonstrated its application in microcirculation mapping in the murine brain. Cortex-wide imaging of stimulus-evoked neuronal activation and subsequently induced hemodynamic responses was achieved at capillary level with subsecond temporal resolution. The proposed approach expands the currently available neuroimaging toolset, facilitating our understanding of complex neural processes, ultimately paving the way for new insights into brain function and its intricate relationship with microcirculation.

## Methods
### Experimental setup
The system diagram is depicted in Fig. 1a. The proposed system follows the typical epi-fluorescence configuration. Two continuous wave (CW) lasers, with wavelengths of 488 nm (Sapphire LPX 488–500, Coherent, USA) and 660 nm (gem 660, Laser Quantum, USA), were implemented to provide GCaMP/DiD two-channel excitation. The laser beams were shaped with a lens pair comprising a convex lens (LA1255-A, EFL = 50 mm, Thorlabs, USA) and an objective (CLS-SL, EFL = 70 mm, Thorlabs, USA) to offer epi-illumination on the sample. The excitation and emission light paths were combined using dichroic mirrors (Di03-R660-t1-25 × 36 and FF495-Di03-25 × 36, Semrock, USA) for two wavelengths, respectively. In the imaging light path, the backscattered fluorescence was collected with the same objective and then divided by a dichroic mirror (DMLP567L, Thorlabs, USA) towards different detection channels. In the GCaMP detection channel, the fluorescence was filtered with a band-pass filter (MF525-39, Semrock, USA), focused with a camera lens (Laowa Venus 60 mm, Laowa, China), and then projected on the EMCCD sensor (Andor Luca R, Oxford Instruments, UK). Similarly, the DiD detection channel incorporates a camera lens (AF micro-Nikkor 60 mm, Nikon, Japan), a long-pass filter (LP02-671RU-25, Semrock, USA), and a high-speed sCMOS camera (pco.dimax S1, PCO AG, Germany). Image acquisition was conducted with Camware software (version 4.12, PCO AG, Germany). The magnification of both channels was initially set to ~0.86. For experiments involving craniotomy, the magnification ratio was increased to 1.5 or 2.75 by employing various combinations of camera lenses. The specific imaging parameters for each experiment are detailed in Supplementary Table 1.

### Animal work
All animal experiments were performed in accordance with the Swiss Federal Act on Animal Protection and approved by the Cantonal Veterinary Office Zurich.

**Animal model.** To evaluate the system performance, we used a GCaMP6f mouse (21 weeks old, female, C57BL/6J-Tg (Thy1-GCaMP6f) GP5.17Dkim/J, the Jackson Laboratory, USA) for the measurement. To validate the imaging depth of RBC-aided WLFM and assess the impact of the number of injected stained RBCs on imaging quality, we imaged a C57BL/6 mouse (19 weeks old, female, Envigo BMS B.V., Netherlands, $n = 1$). For stimulation related experiments, C57BL/6 mouse (38 weeks old, female, Envigo BMS B.V., Netherlands, $n = 1$) and GCaMP6f mice (9–15 weeks old, male, $n = 5$) with craniotomy were measured during hindpaw and/or whisker stimulations. For transcranial demonstration, GCaMP6f mice (6–8 weeks old, male, $n = 4$) with skull intact were imaged during hindpaw stimulation.

The mice were housed in ventilated cages with ad libitum access to food and water. The housing room was kept under controlled conditions: a 12 h dark/light cycle, 22 °C room temperature, and ~50% relative humidity.

**Surgery and anesthesia.** The mice were anesthetized intraperitoneally (i.p.) with ketamine/xylazine (k/x) mixture (100 mg/kg body weight; 10 mg/kg body weight), followed by subcutaneous injection of analgesics (Buprenorphine, 0.1 mg/kg). To prevent potential cardiovascular complications, the initial induction injection was divided into two parts, administered at an interval of 5 min. Throughout the surgery and subsequent recording, the anesthesia was maintained by i.p. injections of the k/x mixture (25 mg/kg body weight; 1.25 mg/kg body weight) every 40–45 min. Under anesthesia, the mouse was immobilized using a customized stereotaxic frame (Narishige International Limited, UK). The body temperate of mouse was maintained at 37 °C with a feedback-regulated heating system (PhysioSuite, Kent Scientific, USA). Simultaneously, other physiological parameters such as respiratory rate, heart rate, peripheral oxygen saturation level were monitored in real-time to assess the anesthesia status.

For transcranial imaging, the scalp was carefully removed to expose the underlying skull. The exposed area was then moisturized using the ultrasound gel (Aquasonic Clear, Parker Laboratories Inc., USA). Procedures involving craniotomy followed the same protocol for anesthesia and scalp removal. Additionally, a 3.5 mm diameter craniotomy was performed over the primary somatosensory cortex using a dental drill (Bien-Air), then sealed with a glass coverslip on top. A catheter filled with phosphate buffered saline (PBS) was inserted in the tail vein for injection before the imaging session.

**Stimulation protocol.** Electrical stimulation was applied in both hindpaw and whisker cases, whereas a pair of electrodes was inserted into the corresponding regions, respectively. The unipolar electrical pulse train (4 Hz frequency, 1 or 5 ms pulse width, 0.5 mA intensity) were generated using a stimulus isolator device (Model A365R, World Precision Instruments, USA) which is externally triggered by a commercial trigger box (PulsePal v2, Sanworks, USA). The pulse width was set to 1 and 5 ms for whisker and hindpaw stimulation, respectively. Each stimulation cycle includes 10 s baseline, 8 s stimulation, and 52 s post stimulation.

**In vivo imaging.** Following the tail vein injection of stained RBC suspension, the image stack was recorded by a high-speed camera operating at a frame rate of 400 to 833 Hz. The camera worked in an external trigger mode, utilizing the same trigger box mentioned above to ensure the synchronization between pre-designed stimulation sequence and recording. Owing to the constraints on data transfer speed between camera and PC, the recording duration for each stimulation cycle was shortened to 30 s to avoid frame loss. A complete imaging session comprised 12 repetitive stimulation cycles.

**Ex vivo validation.** For the validation experiment with 2PM, the mouse was intravenously injected with 150 μL lectin-FITC (Lycopersicon

Esculentum (Tomato) Lectin fluorescein, 2 mg/ml, L32478, Invitrogen, USA) to stain the vasculature with a total circulation time of ~3 min. Following the circulation period, the mouse was euthanized while still under anesthesia. Two-photon imaging was conducted on the extracted mouse brain.

## RBC staining

The RBC staining protocol was adapted from the literature[47]. 40 μL blood was collected from littermates of mice for imaging under anesthesia through the cannulated tail vein. 1% wt/vol Ethylenediaminetetraacetic acid (EDTA, E-5134, Sigma, USA) in 0.7% wt/vol sodium chloride (NaCl, 177410, Fisherscientific, USA) solution was added to the blood as an anticoagulant. The extracted blood was then added into 400 μL Dulbecco's phosphate buffered saline (DPBS, D8537, Sigma-Aldrich, USA), followed by centrifugation at room temperature at 800 r.p.m ($60 \times g$) for 8 min. The supernatant containing plasma and buffy coat was discarded. This procedure was repeated three times. The isolated RBCs were resuspended in 640 μL DPBS, mixed with 640 μL diluent C (CGLDIL, Sigma-Aldrich, USA) and 3.2 μL of DiD (1,1'-Dioctadecyl-3,3,3',3'-Tetramethylindodicarbocyanine, 4-Chlorobenzenesulfonate Salt, D7757, ThermoFisher Scientific, USA) solution at a concentration of 50 mg/mL. The mixture was incubated at 37 °C for 30 s. After incubation, the DiD-stained RBCs were separated from unbounded dye through centrifugation at 800 r.p.m ($60 \times g$) for 8 min, repeated for four times. The stained RBCs were inspected using a commercial confocal microscopy (LSM900, Carl Zeiss AG, Germany). In the end, ~$2 \times 10^7$ stained RBCs were resuspended in 100 μL DPBS for injection. Assuming the total blood volume of a mouse weighing 20 g is 1.5 mL and average RBC density is ~$8.13 \times 10^6/\mu L$ [48,49], the proportion of stained RBCs in relation to the total RBC count is approximately 1.6:1000.

## Two-photon laser scanning microscopy

Two-photon laser scanning microscopy (2PM) was employed to validate the imaging depth of RBC-aided WFLM. The imaging was performed with a custom-built 2PM system[50], equipped with a tunable femtosecond laser (Chameleon Discovery NX, Coherent Inc, USA). In this study, 940 nm excitation wavelength was selected for FITC excitation. A 16× water-immersion objective (CFI75 LWD 16X W, NA = 0.8, Nikon, Japan) was used for the validation experiment. 3D structural imaging was achieved by 2D lateral scanning with galvo mirror and axial scanning of the objective with a piezo motor-driven linear stage at 2 μm step size. Backscattered fluorescence was collected by the same objective, passing through a band-pass filter (535/50 nm, Semrock, USA), and then focused on the photomultiplier tube (H9305-03, Hamamatsu, Japan). Image acquisition was performed with ScanImage (r3.8.1, Janelia Research Campus)[51].

## Localization-based image reconstruction

The process of localization-based image reconstruction, refined based on our previous studies[18,24], involves three main steps: localization, tracking, and superimposition (Supplementary Fig. 1a–d). Each localization image was reconstructed with original frames acquired in 1 s, rendering an effective frame rate of 1 Hz. The localization and tracking of stained RBCs in the image stack were performed using the opensource TrackNTrace toolbox (version 1.03)[52], which was originally developed for single-molecule localization microscopy. An add-on function was implemented to the toolbox, to remove the static background by subtracting the averaged image over the entire recording. Wavelet filtering[53] was used for initial estimates of emitter positions, followed by Gaussian fitting to refine the emitter position with sub-pixel accuracy. After acquiring emitter positions in each frame, the u-track algorithm[54] was employed to acquire particle trajectories by recognizing the same particle in consecutive frames. Three tracking parameters involved include "min-trajectory-length",

"max-track-radius", and "max-frame-gap". The "max-frame-gap" was set to zero so that only particles appearing at consecutive frames were counted. The "min-trajectory-length" set a threshold for the trajectory length. The "max-track-radius" defines the maximum displacement between adjacent frames which was set corresponding to a maximum flow velocity of 20 mm/s. The flow velocity and direction could be determined during the tracking process given the relative displacement of each particle and frame rate. The RBC flux map was rendered by counting the RBC number flowing through each pixel. The velocity and direction maps were calculated by averaging the velocity/direction values deposited in each pixel.

## Data analysis and statistics

**Pulsatility analysis.** Considering the frequency range of blood velocity pulsatility is closely associated with the heart beat which is 300 ~ 450 bpm under anesthesia[55], 20 Hz effective frame rate was chosen to fulfill Nyquist-Shannon sampling criterion to facilitate accurate frequency-domain analysis. The center lines of $n = 50$ artery and vein branches were selected. The velocity of each RBC passing through the selected center lines was recorded over a span of 5 s. Sinusoidal fitting and Fourier transform was performed on the velocity time course to assess pulsatility, using a "sineFit" function (version 3.2.1) available on Mathworks file exchange[56].

**GCaMP activation map and averaged time course calculation.** GCaMP activation map was computed using general linear model (GLM), as detailed in previous studies[16]. In brief, the regressor was calculated by convolving the stimulation paradigm with the GCaMP calcium response function measured with a high-speed camera under the same stimulation. The regressor along with a constant vector was fed into the GLM to estimate the linear coefficient, which formed the final activation map. For each selected ROI (Fig. 3d, k and Fig. 4f), the time course of GCaMP signals was averaged over multiple stimulation cycles. Pearson's correlation coefficients between the measured GCaMP signal time course and the regressor within the stimulation window were calculated using "corrcoef" function with MATLAB (R2020b, MathWorks, USA).

**Velocity activation map calculation.** The stack of velocity maps from an imaging sequence was averaged over multiple stimulation cycles to minimize fluctuations stemming from spontaneous brain activities. Initially, a correlation map was rendered by calculating the Pearson's correlation coefficient between the velocity time course and the boxcar stimulation pattern in a pixelwise way (Supplementary Fig. 9a, b). Pixels exhibiting a correlation coefficient exceeding the 90% quantile were identified and averaged to form a local averaged hemodynamic response curve induced by the stimulation. Subsequently, the final velocity activation map was generated by calculating the pixelwise Pearson's correlation coefficient between the velocity time course and the local averaged hemodynamic response curve (Supplementary Fig. 9c, d).

**Vessel diameter calculation.** The diameters of the selected vessels in this study were defined by the full-width-at-half-maximum (FWHM) of the line profile perpendicular to the vessel central line. The measurements were conducted with the "vessel diameter.ijm" plugin (version 1.0) in ImageJ (version 1.54d, National Institutes of Health, USA)[57].

**Statistics.** All data in this study are presented as the mean ± standard error of the mean (s.e.m). Before conducting statistical analysis, the D'Agostino & Pearson normality test and Shapiro-Wilk test was applied to assess data distribution. Paired $t$-test (two-tailed) was employed for data that passed the normality test, otherwise, non-parametric Wilcoxon test (two-tailed) was utilized instead. For unpaired comparisons, the Mann-Whitney

test (two-tailed) was applied. Statistical methods used are indicated in the figure captions. A P value < 0.05 was recognized as statistically significant. All statistical analyses were performed using GraphPad Prism 9 (v9.5.1, GraphPad Software, USA).

## Reporting summary

Further information on research design is available in the Nature Portfolio Reporting Summary linked to this article.

## Data availability

The main data supporting the finding of this study are available within the main text or Supplementary Information. Source data are provided with this paper. The raw datasets before image reconstruction are too large to be publicly shared, yet they are available for research purposes from the corresponding author upon request. Requests will be fulfilled within 4 weeks. Source data are provided with this paper.

## Code availability

Localization and tracking of fluorescence emitters were performed with the open-source TrackNTrace toolbox[52]. Image reconstruction was performed with custom MATLAB codes which are provided on the Zenodo repository website at: https://doi.org/10.5281/zenodo.10663497.

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

## Acknowledgements

The authors would like to thank M. Reiss for the assistance with animal experimentation. D.R. acknowledges funding from the Swiss National Science Foundation (310030_192757) and the US National Institutes of Health (UF1-NS107680 and R01-NS126102). S.W. acknowledges funding from the Swiss National Science Foundation (310030_200703).

## Author contributions

Z.C. conceived the experimental design. Q.Z., C.G., and Z.C. carried out the experiments. Q.Z. and L.T. performed red blood cell staining. Q.Z., L.G., and Z.C. conducted data analysis and visualization. B.W., S.W., M.E.A., and J.D. contributed to the interpretation of the results. D.R. and Z.C. supervised the work. Q.Z., C.G., and Z.C. wrote the manuscript. All authors reviewed and edited the manuscript.

## Competing interests

The authors declare no competing interests.
