## [Peer Review File · Nature Communications]

Reviewers' Comments:

Reviewer #1:

Remarks to the Author:

This manuscript reports an interesting method with a simple and very accessible set of optics together with cameras to analyze the neurovascular dynamics in behaving mice. The authors first illustrated the RBC-aided imaging design, then validated the design with analyses of vessel structure and hemodynamics. The authors then analyzed the flow dynamics when hind paw stimulation was performed. Neuronal activity and vascular dynamics were simultaneously analyzed during the hind paw and whisker stimulation. Lastly the authors analyzed the neurovascular activity in cortex wide manner. The design has been validated by a set of tests, suggesting the feasibility of this design. Both spatial and temporal resolution of this system are suitable for studying neurovascular dynamics. This method is a great alternative for analyzing brain hemodynamics and neuronal activity simultaneously with a nice spatial and temporal resolution. I support a publication if the following concerns can be addressed.

Major points

1 Figure 1 The dominant frequency for vein shown in Figure 1g is ambiguous. The diameter of artery and vein should be provided for a better understanding of the ambiguous frequency for the vein. The RBC dynamics may be affected by the vessel diameters. This raised a general concern whether the diameters of either artery or vein affect the analyses.

2 In figure 2f, a statistical analysis should be provided. Which one is statistically elevated? When was it significantly elevated? Relatedly, if statistically, do we expect artery velocity to be elevated before the capillary or vein? Do we expect RBC flux in artery is larger than those of capillary and vein. Without a set of comprehensive analyses, it is hard to appreciate whether this method is sufficient to measure these changes. Similar to the concern in Figure 1, it remain unknown whether the diameter of either artery or vein affects the analyses?

Also related to this figure, there is a description "we observed that vessel diameter change reached its peak significantly earlier (4.17 ± 0.53 s) than that of velocity change (8.92 ± 0.58 s) in arteries (Fig. 2f, Supplementary Fig. 1), which indicates vasodilation in arteries plays a more important role in regulating the blood flow volume upon external stimulation". This can be interesting, but can this conclusion also be drawn from analyses of an individual vessel? Otherwise, it is so hard to make this conclusion with the traces presented in figure 2f. Importantly, the data seems to represent vessels with different diameters.

3 For figures 3 and 4, the concern is similar to that of Figures 1 and 2. The traces in Figures 3d,e,g,k,l and n should be statistically analyzed. With current plots, it is rather hard to tell whether there is a change or not.

Minor

1, the definition of artery and vein should be clearly described in the very beginning.

2, In Figure 2, which brain area were the images from?

3, the temporal resolution is 1 s. The authors should clearly describe this in the result.

Reviewer #2:

Remarks to the Author:

The authors introduced a localization microscopy with fluorescently sparsely labeled red blood cells, achieving a cortex-wide imaging scale, 4.9- μ m spatial resolution and 20 Hz temporal resolution. The idea that applying single molecular localization in the in vivo dynamical observation is quite interesting. However, there are some unclear points in the article.

1. What is the optical setup (NA, magnification, spatial sampling rate) of the imaging system? This information is missing in the text. The authors claimed that the theoretical resolution of localization-based imaging methods primarily hinges on the SNR, while the reviewer thinks if undersampling is too severe, the accuracy of reconstruction will be greatly affected.

2. What is the imaging depth of the system? How does the scattering of tissue affect the imaging quality?

3. RBCs go slowly through tiny capillaries, won't it cause reconstruction discontinuity on those capillaries using the method in this paper? Or it should take much more frames to reconstruct than large vessels?

4. How did authors determine that an effective reconstruction should use 20 frames instead of

other numbers?

5. How to optimize the concentration of injected RBCs? The authors should compare reconstructed imaging quality using various fluorescent RBC concentrations.

6. Tao et al. have developed a RUSH technique (Nat. Photonics 13, 809–816 (2019)) that can imaging with a centimeter-field of view, 1.2- μm resolution at 30 fps. The authors should compare their new imaging system with RUSH.

7. Photoacoustic microscopy could also obtain vascular structure, blood flow velocity, even blood oxygen saturation with promising spatial-temporal resolution. The authors are also suggested to discuss in the paper.

8. Instead of open-skull window, novel optical clearing skull windows (Light-Science & Applications , 2018, 7 (2) : 17153; eLight, 2022, 2, 15) are also capable of high-resolution cortical imaging. Does this localization microscopy suitable for optical clearing skull windows?

Reviewer #3:

Remarks to the Author:

This manuscript introduces the RBC-aided widefield localization microscopy (WFLM) technique and demonstrates its performance in mapping microcirculation in the murine brain. The authors showcase the methodology and its imaging performance. Then, the authors validated the system's capability of sensory stimulation-evoked hemodynamic responses at the capillary level.

Subsequently, the authors demonstrated the integration of the system with epifluorescence calcium imaging to record the relationship between neural activity and hemodynamic responses. Finally, the authors exploited the method's ability for cortex-wide transcranial imaging. This work provides a valuable imaging tool for studying neurovascular coupling. However, WFLM, RBC staining, and epifluorescence calcium imaging are well-established techniques. Therefore, from an innovative perspective, the manuscript seems to lack some novelty. Therefore, I recommend that this article may not be suitable for publication in Nature Communication. Please see my concerns below.

1. My major concern is whether the proposed cortex-wide transcranial imaging can be used to record blood flow responses at capillary resolution. Most of the experiments in the manuscript are conducted through a cranial window for high-resolution imaging, and transcranial imaging of the entire cortex seems to have lower resolution, capturing only the blood flow responses of larger arteries and veins.

2. How are arteries and veins distinguished? What is the meaning of the direction maps in Supplementary Figure 2? Why is the blood flow direction inconsistent within the same vessel?

3. The Results section includes four components. The authors used camera lenses with different magnification ratios, and it would be beneficial to clearly specify the imaging resolution and FOV corresponding to each experiment. In the 'system performance' section (Result 1), the resolution is stated as 4.9 μm . Is the resolution consistent with the capillary-level multiparametric mapping experiment (Result 2)? In transcranial imaging (Result 4), the presence of the skull introduces uneven scattering, leading to a decrease in SNR. Under these conditions, what is the spatial resolution of the imaging, and can it be used to monitor changes in blood flow parameters in capillaries?"

4. The authors directly presented the results of the localization-reconstructed structural map in the results section. In the supplementary materials, it would be beneficial to showcase consecutive single-frame RBC labeling images and provide an explanation of the reconstruction process for intensity maps and velocity maps.

5. In line 96, "The sparsity of stained RBCs in the total blood can be controlled by adjusting the number of stained RBCs for injection." So, what is the relationship between the sparsity of stained RBCs, magnification, and frame rate?

6. What is the temporal and spatial resolution of calcium imaging? In the results, only GCaMP activation maps are shown. Could it be possible to dynamically present the changes in GCaMP

signals and blood flow parameter signals in the form of a movie?

7. In result 2, activation areas were detected using ISOI, while in result 3, epifluorescence calcium imaging was used to provide activation areas. Both experiments recorded hemodynamic responses after hindpaw stimulation. What is the difference between the two experiments?

8. In Fig 3 g,n, what does the purple dashed line represent? The author provided the velocity time courses in different segments of the venous vascular tree, suggesting that smaller vessels exhibit more pronounced changes in velocity. Could the author provide the changes in velocity for the same vascular tree under stimulated and non-stimulated conditions? Additionally, how about the changes in the arterial vascular tree?"

9. In Fig 1, velocity information for two vessels (artery and vein) is provided. Can the velocity of capillaries be recorded? What are the differences compared to arteries and veins?

10. In the introduction section, a comparison should be made between laser speckle contrast imaging and optical coherence tomography angiography. Additionally, in vivo skull optical clearing technique, combined with various optical imaging methods, enables monitoring of blood flow and neural signals in live mice, even achieving 3D imaging. What are the advantages and disadvantages of the proposed method compared to these techniques?

Point-by-Point Response to the Reviewers' comments

Reviewer #1

This manuscript reports an interesting method with a simple and very accessible set of optics together with cameras to analyze the neurovascular dynamics in behaving mice. The authors first illustrated the RBC-aided imaging design, then validated the design with analyses of vessel structure and hemodynamics. The authors then analyzed the flow dynamics when hind paw stimulation was performed. Neuronal activity and vascular dynamics were simultaneously analyzed during the hind paw and whisker stimulation. Lastly the authors analyzed the neurovascular activity in cortex wide manner. The design has been validated by a set of tests, suggesting the feasibility of this design. Both spatial and temporal resolution of this system are suitable for studying neurovascular dynamics. This method is a great alternative for analyzing brain hemodynamics and neuronal activity simultaneously with a nice spatial and temporal resolution. I support a publication if the following concerns can be addressed.

Reply: We appreciate the reviewer for recognizing the value of our work. Please find below our point-by-point answers to the comments.

Major points

1. Figure 1 The dominant frequency for vein shown in Figure 1g is ambiguous. The diameter of artery and vein should be provided for a better understanding of the ambiguous frequency for the vein. The RBC dynamics may be affected by the vessel diameters. This raised a general concern whether the diameters of either artery or vein affect the analyses.

Reply: We thank the reviewer for this insightful suggestion. The diameters of representative vessels have been added in the revised Fig. 1g. The larger diameter of the selected vein may contribute to the ambiguous dominant frequency in veins, whereas a similar phenomenon was observed in the literature¹.

To investigate whether vessel diameter influences pulsatility analysis, we analyzed $n = 50$ vessel branches with diameters ranging from 12.6 to 119.0 μm , including $n = 29$ arteries and $n = 21$ veins. The calculated dominant frequency is 4.61 ± 0.03 Hz (mean \pm s.e.m), demonstrating a good consistency across vessels of varying diameters (Fig. 1i). Moreover, the calculated Pearson's correlation coefficient of 0.11 indicates a weak correlation between the dominant frequency of velocity fluctuations and vessel diameter, which corroborates the capability of RBC-WFLM in accurately quantifying the flow dynamics across different vessel compartments (Fig. 1i).

The corresponding descriptions have been added to "Results – Methodology of RBC-aided WFLM and characterization" section (page 3, paragraph 3). The method for vessel diameter measurement has been added to "Methods – Data analysis and statistics" section (page 10, paragraph 4).

References:

1 Meng, G. *et al.* Ultrafast two-photon fluorescence imaging of cerebral blood circulation in the mouse brain in vivo. *Proceedings of the National Academy of Sciences* **119**, e2117346119 (2022).

2. In figure 2f, a statistical analysis should be provided. Which one is statistically elevated? When was it significantly elevated? Relatedly, if statistically, do we expect artery velocity to be elevated before the capillary or vein? Do we expect RBC flux in artery is larger than those of capillary and vein. Without a set of comprehensive analyses, it is hard to appreciate whether this method is sufficient to measure these changes. Similar to the concern in Figure 1, it remain unknown whether the diameter of either artery or vein affects the analyses?

Also related to this figure, there is a description "we observed that vessel diameter change reached its peak significantly earlier (4.17 ± 0.53 s) than that of velocity change (8.92 ± 0.58 s) in arteries (Fig. 2f,

Supplementary Fig. 1), which indicates vasodilation in arteries plays a more important role in regulating the blood flow volume upon external stimulation”. This can be interesting, but can this conclusion also be drawn from analyses of an individual vessel? Otherwise, it is so hard to make this conclusion with the traces presented in figure 2f. Importantly, the data seems to represent vessels with different diameters.

Reply: We thank the Reviewer for the constructive suggestion. For each response curve in Fig. 2f, we performed statistical analysis on the velocity, diameter, RBC flux between the baseline and response peak time point. The results are shown in supplementary Figs. 4a, b, which confirm that all 9 curves in Fig. 2f exhibit a significant increase upon stimulation. For statistics, two-sided paired t-test was applied to datasets that passed the D'Agostino & Pearson normality test; otherwise, a two-sided paired Wilcoxon test was utilized instead.

To investigate the difference in response parameters across vessel types, we selected $n = 20$ arteries, $n = 10$ veins, and $n = 5$ capillaries for further analysis in the revised supplementary Fig. 4. We quantified the response time by calculating the time-to-peak (TTP) value for both flow velocity and vessel diameter response curves. Our analysis revealed the TTP for velocity response is longer in arteries than in veins and capillaries, whereas the TTP for diameter response is shorter in arteries than in veins and capillaries (supplementary Fig. 4e). Additionally, in terms of velocity and RBC flux, capillaries exhibited a higher fractional change in both parameters compared to veins and arteries (supplementary Fig. 4c). By analyzing multiple vessels across all three vessel types, we have validated the capability of RBC-aided WFLM in quantifying stimulation triggered hemodynamic responses across various vessel types and diameters. The diameters of selected vessels have been added to Fig. 2f.

Concerning the TTP of vessel diameter and velocity change in arteries, the response curve in Fig. 2f and supplementary Fig. 4d is collected from a representative artery. We also extended this analysis to include $n = 20$ arteries of varying diameters where the same phenomenon was confirmed with a statistical analysis (supplementary Fig. 4e).

The corresponding results and descriptions have been added to supplementary Fig. 4 and “Results – Capillary-level multiparametric mapping of sensory stimulation-evoked hemodynamic responses” section (page 4, paragraph 1), respectively. The method used for statistical analysis has been added to “Methods – Data analysis and statistics” section (page 10, paragraph 5).

3. For figures 3 and 4, the concern is similar to that of Figures 1 and 2. The traces in Figures 3d,e,g,k,l and n should be statistically analyzed. With current plots, it is rather hard to tell whether there is a change or not.

Reply: As suggested, we conducted a statistical analysis on the GCaMP time courses in Figs. 3d, k by calculating the Pearson's correlation coefficient (r) and corresponding P value between measured GCaMP time course and the regressor in the stimulation window. The regressor was generated by convolving the stimulation pattern and GCaMP calcium response function as described in the Methods section. The method to calculate Pearson's correlation coefficient and P value has been added to “Methods - Data analysis and statistics” section (page 10, paragraph 2). In both hindpaw and whisker stimulation cases, the r value is higher in the regions of interest (ROIs) inside GCaMP activation region than those outside GCaMP activation region. The determined r and p values have been incorporated into the revised Figs. 3d, k.

Additionally, we performed a statistical analysis of velocity time courses (Fig. 3e, g, l, n) by comparing the flow velocity between the baseline and response peak time point. The supplementary Fig. 5 and supplementary Fig. 6 have been expanded to include the results of this statistical analysis. In the case of hindpaw stimulation, all selected vessels inside GCaMP activation region, 3 out of 5 vessels outside GCaMP activation region, and vessels along the selected venule tree exhibit a significant increase in velocity upon stimulation (supplementary Figs. 5d-f). Similar results were observed in whisker stimulation case (supplementary Figs. 6d-f).

Similarly, statistical analysis results of GCaMP and velocity time courses have been added to Fig. 4f and supplementary Fig. 7.

Minor

1. the definition of artery and vein should be clearly described in the very beginning.

Reply: We thank the Reviewer for this suggestion. The differentiation between arteries and veins in our study is based on their distinct flow directions, which could be discerned from flow direction map. Moreover, we included time-lapse images of a representative artery and vein presenting inverse flow directions in the supplementary Figs. 1e, f as an example. The relevant descriptions have been incorporated into “Results - Methodology of RBC-aided WFLM and characterization” section (page 3, paragraph 2).

2. In Figure 2, which brain area were the images from?

Reply: The images in Fig. 2 were taken from primary somatosensory cortex. To make it clear, we added a schematic in the revised Fig. 2a to label the brain area imaged. In addition, we added corresponding descriptions in “Methods – Animal” section (page 8, paragraph 4).

3. the temporal resolution is 1 s. The authors should clearly describe this in the result.

Reply: As suggested, we added the description on temporal resolution in “Result – Capillary-level multiparametric mapping of sensory stimulation-evoked hemodynamic responses” section (page 3, paragraph 4).

Reviewer #2

The authors introduced a localization microscopy with fluorescently sparsely labeled red blood cells, achieving a cortex-wide imaging scale, 4.9- μm spatial resolution and 20 Hz temporal resolution. The idea that applying single molecular localization in the in vivo dynamical observation is quite interesting. However, there are some unclear points in the article.

Reply: We thank the Reviewer for the positive comments and constructive suggestions. Please find below our answers to the specific comments.

1. What is the optical setup (NA, magnification, spatial sampling rate) of the imaging system? This information is missing in the text. The authors claimed that the theoretical resolution of localization-based imaging methods primarily hinges on the SNR, while the reviewer thinks if undersampling is too severe, the accuracy of reconstruction will be greatly affected.

Reply: We thank the reviewer for pointing out this issue. In this study, we adjusted the combination of objective and tube lens for different experiments to balance the spatial sampling rate and field of view (FOV). To make it clear, we added a supplementary Table 1 (Supplementary Information, page 10) to demonstrate the specifications (i.e., frame rate, magnification, NA, FOV, pixel size etc.) of imaging system for each experiment.

As the reviewer correctly pointed out, the theoretical resolution of localization-based imaging not only depends on SNR but also influenced by the spatial sampling rate as demonstrated in super-resolution microscopy¹. Thus, the relevant sentence in the main text was amended to “Note that theoretical resolution of localization-based imaging methods primarily hinges on the signal-to-noise ratio (SNR) and spatial sampling rate of the system” in the “Results - Methodology of RBC-aided WFLM and characterization” section (page 3, paragraph 2).

References:

1. Thompson, R. E., Larson, D. R. & Webb, W. W. Precise nanometer localization analysis for individual fluorescent probes. *Biophysical journal* **82**, 2775-2783 (2002).

2. What is the imaging depth of the system? How does the scattering of tissue affect the imaging quality?

Reply: To quantify the imaging depth of the proposed system, we performed validation experiment by imaging the same mouse brain with both RBC-aided WFLM and two-photon laser scanning microscopy (2PM). The mouse with cranial window implanted was firstly injected with stained RBCs to acquire the localization image, followed by the intravenous injection of FITC-lectin to stain the vasculature. We imaged the extracted mouse brain *ex vivo* with 2PM using a water-immersion 16 \times objective and under 940 nm excitation wavelength. 3D image stacks were collected with additional axial scanning at a 2 μm step size, where color-encoded depth maps were rendered. After image co-registration of the same ROIs collected with both RBC-aided WFLM and 2PM, we selected 10 vessels with their depths estimated from 2PM. The deepest vessel detected by RBC-aided WFLM is located \sim 200 μm below the brain surface. The relevant results and descriptions were added to supplementary Fig. 3 and “Results - Methodology of RBC-aided WFLM and characterization” section (page 3, paragraph 2). The details for animal experiments were added to “Methods – Animal” (page 8, paragraph 1) and “Methods – Two-photon laser scanning microscopy” sections (page 9, paragraph 2).

The impact of scattering on image quality comes from two main aspects. Firstly, the SNR is expected to decrease with the scattering, directly affecting both the localization accuracy and achievable imaging depth. Secondly, the random scattering caused by the inhomogeneous biological samples may lead to distortions in the point spread function (PSF), thus degrading localization accuracy with Gaussian fitting-based methods. We added a discussion on this point in the revised manuscript (page 6, paragraph 3).

3. RBCs go slowly through tiny capillaries, won't it cause reconstruction discontinuity on those capillaries using the method in this paper? Or it should take much more frames to reconstruct than large vessels?

Reply: We thank the reviewer for this insightful question. Indeed, the RBC flux in capillaries is relatively lower than in arteries and veins. Consequently, a larger number of original frames is required to accurately reconstruct the structure and to analyze the flow dynamics within capillaries. To illustrate this point, we incorporated the velocity time course from a representative capillary in Fig. 1g. The measured flow velocity in the capillary was much lower than those in vein and artery. To avoid the empty pixels in the capillary, we opted to reconstruct the image at an effective frame rate at 1 Hz (i.e., employing 400 frames for reconstruction), in contrast to 20 Hz frame rate for the vein and artery. Consequently, in the stimulation-related experiments in the original manuscript, we set the temporal resolution to 1 s to facilitate the detection of hemodynamic responses across various vessel types. The relevant descriptions have been added to "Results - Methodology of RBC-aided WFLM and characterization" section (page 3, paragraph 3).

4. How did authors determine that an effective reconstruction should use 20 frames instead of other numbers?

Reply: The choice to employ a temporal resolution of 20 Hz (20 frames for reconstruction) in Fig. 1 was determined by the need to capture the pulsatile flow, which is expected to closely align with the heartbeat of approximately 300-450 bpm under anesthesia (corresponding to 5-7.5 Hz frequency range). To adequately observe the pulsatile flow within this frequency range, we selected an effective frame rate of 20 Hz to fulfill Nyquist-Shannon sampling criterion. A higher frame rate is not used to avoid the empty pixels in the selected artery and vein. We have expanded "Methods - Data analysis and statistics - Pulsatility analysis" to include these descriptions (page 10, paragraph 1).

5. How to optimize the concentration of injected RBCs? The authors should compare reconstructed imaging quality using various fluorescent RBC concentrations.

Reply: We thank the Reviewer for this insightful question. Indeed, the quantity of injected stained RBCs is expected to significantly influence the WFLM image quality. In the ideal case, a larger number of injected stained RBCs is expected to provide more resolvable structures at a given integration time. To illustrate this effect, we performed a new experiment to compare the reconstructed image quality versus fluorescently-labeled RBC concentrations. The result has been added in the new supplementary Fig. 2, where the localization image of the same mouse was reconstructed post intravenous injection of three different quantities of stained RBCs, i.e., 1×10^6 , 1×10^7 , 2×10^7 . With the fixed frame rate, as the number of stained RBCs increases, more vascular structures were detected in the localization image, attributed to the higher number of detected stained RBCs. However, a higher density of stained RBCs within the single frame may compromise localization accuracy due to potential PSF overlapping and increase the likelihood of tracking errors. Considering these factors, in this study, we opted for 2×10^7 stained RBCs for injection to balance the image quality and localization and tracking accuracy. The optimal quantity of injected RBCs needs to be adapted under different imaging systems, considering the resolution, magnification, and frame rate.

6. Tao et al. have developed a RUSH technique (Nat. Photonics 13, 809-816 (2019)) that can imaging with a centimeter-field of view, 1.2- μ m resolution at 30 fps. The authors should compare their new imaging system with RUSH.

Reply: We thank the Reviewer for mentioning this work. RUSH technique was proposed to overcome the constraint of space-bandwidth product (SBP) in conventional optical microscopy, which achieved a relatively uniform spatial resolution of $\sim 1.2 \mu\text{m}$ across centimeter-scale FOV at video rate, enabling cortex-wide tracking of immune cells and vascular dynamic imaging in mice.

Here, we listed several points for comparison between our proposed method and RUSH:

Optical design: RBC-aided WFLM is grounded on the classical widefield excitation/detection paradigm, utilizing a simple and cost-effective optical design and off-the-shelf optical components. RUSH employed 35 sCMOS cameras in conjunction with a redesigned microscopic objective and a complex data management system.

Resolution and FOV: While RBC-aided WFLM and RUSH aim to enhance the lateral resolution across a large FOV, different strategies were utilized in these two methods. The resolution improvement of RUSH benefits from its unique “flat-curved-flat” imaging configuration with a high-NA customized objective (NA: 0.35). In contrast, RBC-aided WFLM operates with a low NA system (NA: 0.04-0.08), with resolution improvement beyond the objective-defined diffraction limit primarily derived from the sub-pixel localization process.

Frame rate: RUSH is featured with 30 Hz effective frame rate for cortex-wide tracking of immune cells and vascular dynamic imaging in mice. In contrast, RBC-aided WFLM exploited an original frame rate of 400-833 Hz to enable the tracking of rapid flow dynamics in vascular network with flow velocity up to 20 mm/s.

Imaging condition: RBC-aided WFLM is showcased the feasibility of transcranial imaging, which is still challenging for most fluorescence imaging methods with high-NA designs.

The relevant descriptions and references have been added to “Discussion” section (page 6, paragraph 3).

7. Photoacoustic microscopy could also obtain vascular structure, blood flow velocity, even blood oxygen saturation with promising spatial-temporal resolution. The authors are also suggested to discuss in the paper.

Reply: As recommended, we have expanded the discussion on photoacoustic/optoacoustic microscopy and its advantages and limitations in vascular imaging. The relevant descriptions and references have been added to “Discussion” section (page 6, paragraph 4).

8. Instead of open-skull window, novel optical clearing skull windows (Light-Science & Applications, 2018, 7(2): 17153; eLight, 2022, 2, 15) are also capable of high-resolution cortical imaging. Does this localization microscopy suitable for optical clearing skull windows?

Reply: We thank the Reviewer for this inspiring question. The reported skull optical clearing method holds promise to be incorporated with RBC-aided WFLM to mitigate the effects of skull-induced scattering on imaging performance, given its demonstrated application in multi-photon imaging. The corresponding descriptions and references have been added to “Discussion” section (page 6, paragraph 3).

Reviewer #3

This manuscript introduces the RBC-aided widefield localization microscopy (WFLM) technique and demonstrates its performance in mapping microcirculation in the murine brain. The authors showcase the methodology and its imaging performance. Then, the authors validated the system's capability of sensory stimulation-evoked hemodynamic responses at the capillary level. Subsequently, the authors demonstrated the integration of the system with epifluorescence calcium imaging to record the relationship between neural activity and hemodynamic responses. Finally, the authors exploited the method's ability for cortex-wide transcranial imaging. This work provides a valuable imaging tool for studying neurovascular coupling. However, WFLM, RBC staining, and epifluorescence calcium imaging are well-established techniques. Therefore, from an innovative perspective, the manuscript seems to lack some novelty. Therefore, I recommend that this article may not be suitable for publication in Nature Communication. Please see my concerns below.

Reply: We thank the Reviewer for recognizing the value of our work along with the comprehensive feedback provided. Regarding the novelty, despite that WFLM utilizing fluorescent beads as targets for localization has been demonstrated in our previous work¹, it suffers from a low temporal resolution (several minutes to reconstruct an image) due to the short circulation time of fluorescent beads. This limitation restricts its application in functional neuroimaging, let alone the potential biosafety issues using polyethylene material. In the current work, owing to the long circulation time and superior biocompatibility of stained RBCs, the temporal resolution has been boosted to 20 Hz for arteries and veins and 1 Hz for capillaries across the entire cortex, not accessible in other localization-based imaging techniques. More importantly, when combined with epifluorescence calcium imaging, it offers a comprehensive set of multiparametric information for unraveling intricate neurovascular coupling mechanisms. This provides a versatile toolset for addressing fundamental questions in neuroscience and investigating disease pathologies. Please find our point-by-point responses below.

References:

1. Chen, Z., Zhou, Q., Robin, J. & Razansky, D. Widefield fluorescence localization microscopy for transcranial imaging of cortical perfusion with capillary resolution. *Optics letters* **45**, 3470-3473 (2020).

1. My major concern is whether the proposed cortex-wide transcranial imaging can be used to record blood flow responses at capillary resolution. Most of the experiments in the manuscript are conducted through a cranial window for high-resolution imaging, and transcranial imaging of the entire cortex seems to have lower resolution, capturing only the blood flow responses of larger arteries and veins.

Reply: As the Reviewer correctly pointed out, the presence of the skull introduces additional scattering, compromising the SNR and thus affecting the localization accuracy. To assess the practical imaging resolution under transcranial case, we checked the full-width-at-half-maximum (FWHM) of some discernible small vessels. The calculated resolution of transcranial imaging with RBC-aided WFLM was 9.92 μm . We added this value along with system configurations to supplementary Table 1 (Supplementary Information, page 10). Accordingly, we expanded the "Discussion" section to discuss the impact of skull scattering on the achievable spatial resolution (page 6, paragraph 3).

Despite that the achieved resolution could support the structural imaging of capillaries (typically $<10 \mu\text{m}$), it is challenging to measure the hemodynamic responses in capillaries given the fact that there are much less RBCs in capillaries compared to larger arteries and veins with diameters of $\sim 15 \mu\text{m}$ where hemodynamic responses can still be reliably detected. These results and corresponding descriptions have been added to supplementary Fig. 8 and "Results – Transcranial imaging of cortex-wide neurovascular activation" section (page 5, paragraph 3).

2. How are arteries and veins distinguished? What is the meaning of the direction maps in Supplementary Figure 2? Why is the blood flow direction inconsistent within the same vessel?

Reply: In this work, we mainly distinguish artery and vein based on their distinctive flow directions and prior knowledge of typical pial vessels including anterior cerebral arteries (ACAs), middle cerebral arteries (MCAs), and drainage veins. To better illustrate this point, we provided the time-lapse images of an artery and vein in the new supplementary Figs. 1e, f where opposite flow directions of artery and vein were depicted.

The orientation in flow direction maps is encoded from 0 to 360° with six colors (each color represents 60° direction range). To improve the interpretation on flow direction, we replaced the previous color bar with a color wheel (Fig. 1e and supplementary Fig. 5 and 6) by labeling the starting direction (0°) and ending direction (360°). Since the direction of velocity is defined by the angle relative to the starting direction instead of vessel center line, the blood flow direction will be regulated by the vessel tortuosity and can be discontinued if shown with a low number of orientations (i.e., 6 orientations within 360° in this work).

3. The Results section includes four components. The authors used camera lenses with different magnification ratios, and it would be beneficial to clearly specify the imaging resolution and FOV corresponding to each experiment. In the 'system performance' section (Result 1), the resolution is stated as 4.9 μm. Is the resolution consistent with the capillary-level multiparametric mapping experiment (Result 2)? In transcranial imaging (Result 4), the presence of the skull introduces uneven scattering, leading to a decrease in SNR. Under these conditions, what is the spatial resolution of the imaging, and can it be used to monitor changes in blood flow parameters in capillaries?"

Reply: We thank the reviewer for the suggestion. In the updated manuscript, we have added the specifications of imaging systems (i.e., frame rate, magnification, NA, FOV, pixel size etc.) for each experiment in the supplementary Table 1 (Supplementary Information, page 10).

The calculated resolution has been also added in supplementary Table 1, which is 4.90 μm, 2.75 μm, 4.90 μm, and 9.92 μm for Results 1 to 4, respectively. The achievable resolution in experiments utilizing a cranial window (Result 1 to 3) is sufficient for the observation of hemodynamic changes in capillaries, as demonstrated in Figs. 2 and 3. In transcranial imaging case, the degraded SNR complicates detection of sufficient stained RBCs to accurately depict hemodynamic responses in capillaries. However, our method retains the capacity to detect activations in arteries and veins with diameters as small as ~15 μm (supplementary Fig. 8).

4. The authors directly presented the results of the localization-reconstructed structural map in the results section. In the supplementary materials, it would be beneficial to showcase consecutive single-frame RBC labeling images and provide an explanation of the reconstruction process for intensity maps and velocity maps.

Reply: We thank the reviewer for this suggestion. A reconstruction pipeline including consecutive single frames was provided in supplementary Fig 1.

5. In line 96, "The sparsity of stained RBCs in the total blood can be controlled by adjusting the number of stained RBCs for injection." So, what is the relationship between the sparsity of stained RBCs, magnification, and frame rate?

Reply: We apologize for any confusion caused. The sparsity of stained RBCs in the total blood is primarily determined by the ratio of injected stained RBCs to the total RBC count, which is independent of the specifications of the imaging system. From the imaging perspective, the accuracy of localization and tracking is influenced by magnification and frame rate of the imaging system. Thus, we have amended this sentence to "The sparsity of stained RBCs in the total blood can be controlled by adjusting the number of stained RBCs for injection, while the localization and tracking accuracy is influenced by magnification and frame rate of the imaging system" (page 3, paragraph 2).

6. What is the temporal and spatial resolution of calcium imaging? In the results, only GCaMP activation maps are shown. Could it be possible to dynamically present the changes in GCaMP signals and blood flow parameter signals in the form of a movie?

Reply: The temporal resolution of calcium imaging is 40 Hz in Result 3 and 20 Hz in Result 4. The theoretical spatial resolution was kept the same as 18.60 μm in both experiments. The specifications of epifluorescence calcium imaging system have been added to supplementary Table 1 (supplementary information, page 10).

In addition, we have added supplementary Video 2 to present the changes of GCaMP signal and velocity in the mouse brain post whisker stimulation.

Supplementary Table 1. Imaging conditions for experiments.

	RBC-aided WFLM						Epi-fluorescence calcium imaging	
	Frame rate (Hz)	Magnification ratio	NA	FOV (mm^2)	Pixel size (μm)	FWHM of the smallest vessel structure (μm)	Frame rate (Hz)	Resolution (μm)
Fig. 1	400	1.50	0.04	7.39×7.39	7.33	4.90	No calcium recording performed	
Fig. 2	833	2.75	0.08	4.03×4.03	4.00	2.75	No calcium recording performed	
Fig. 3	400	1.50	0.04	7.39×7.39	7.33	4.90	40	18.60
Fig. 4	400	0.86	0.04	12.94×12.94	12.83	9.92	20	18.60

Abbreviations: NA, numerical aperture. FOV, field of view; FWHM, full-width-at-half-maximum.

7. In result 2, activation areas were detected using ISOI, while in result 3, epifluorescence calcium imaging was used to provide activation areas. Both experiments recorded hemodynamic responses after hindpaw stimulation. What is the difference between the two experiments?

Reply: The specifications of imaging system for Result 2 and 3 were different. Result 2 is featured with a larger magnification ratio and higher frame rate as shown in supplementary Table 1, resulting in a reduced FOV and enhanced spatial resolution compared to Result 3. One wildtype C57BL/6 mouse was used in Result 2 without GCaMP expression, thus we utilized ISOI to identify activation areas before performing the RBC-aided WFLM.

In Result 3, the focus was shifted to demonstrate the multimodal possibilities of RBC-aided WFLM, by incorporating epifluorescence calcium imaging. This combination enabled concurrent observation of neuronal calcium signals and induced hemodynamic responses in the transgenic Thy1-GCaMP6f mouse model. In addition, the GCaMP readings provide the neural activation, representing complementary information to the hemodynamic activation captured by RBC-aided WFLM.

8. In Fig 3g,n, what does the purple dashed line represent? The author provided the velocity time courses in different segments of the venous vascular tree, suggesting that smaller vessels exhibit more pronounced changes in velocity. Could the author provide the changes in velocity for the same vascular tree under stimulated and non-stimulated conditions? Additionally, how about the changes in the arterial vascular tree?"

Reply: The dashed purple line in the original Fig. 3g, n indicates the latency of the velocity changes to reach their peaks from different vessel branches. To avoid any confusion, we used “+” sign to mark the time point when the velocity change reaches its peak in the revised Fig. 3g, n to highlight that high-order branch were pronounced at earlier time points.

As suggested, we calculated the velocity change of vessel branches along the venule tree in the baseline and stimulation period in the new supplementary Fig. 5f and supplementary Fig. 6f, together with statistical analysis. In addition, we added the response curves of vessel segments selected from arterial vascular tree in the new supplementary Figs. 5g, h and supplementary Figs. 6g, h where we also observed high-order vessel branches are featured with higher fractional changes of velocity. Since pial artery has fewer branches on the surface compared to veins, 3 and 4 branches were selected from two representative arterial trees compared to 6 branches in the venule tree.

9. In Fig 1, velocity information for two vessels (artery and vein) is provided. Can the velocity of capillaries be recorded? What are the differences compared to arteries and veins?

Reply: The capillary velocity can be recorded as well. However, the RBC flux in capillary is relatively low compared to larger arteries and veins, making it challenging to record capillary velocity at 20 Hz frame rate. Thus, we showcased the capillary velocity at a temporal resolution of 1 s (Fig. 1g). Compared to the artery and vein selected, the measured velocity is much lower as expected. The relevant descriptions have been added to “Results - Methodology of RBC-aided WFLM and characterization” section (page 3, paragraph 3).

10. In the introduction section, a comparison should be made between laser speckle contrast imaging and optical coherence tomography angiography. Additionally, in vivo skull optical clearing technique, combined with various optical imaging methods, enables monitoring of blood flow and neural signals in live mice, even achieving 3D imaging. What are the advantages and disadvantages of the proposed method compared to these techniques?

Reply: We thank the reviewer for this inspiring suggestion.

Compared to laser speckle contrast imaging (LSCI), our method is featured with much enhanced lateral resolution benefiting from super-resolution localization concept, which make it possible for us to readout hemodynamic readings at a single-vessel level. From the functional imaging perspective, it is still challenging for LSCI to acquire quantitative blood flow parameters (e.g., flow velocity, RBC flux) due to the equivocal model of speckle contrast¹. In contrast, RBC-aided WFLM provided quantitative readings of flow velocity/flux/vessel diameter.

Compared to optical coherence tomography (OCT), RBC-aided WFLM is lack of axial resolvability. Both RBC-aided WFLM and OCT can simultaneously measure velocities in multiple vessels. However, OCT provides the velocity map from repetitive B-scans which renders velocity maps in either coronal or sagittal view. In contrast, RBC-aided WFLM provided a transverse view of the entire pial network. So far, applying OCT for stimulus evoked brain activation mapping is not widely exploited probably due to the limited scanning speed and FOV². As suggested, we expanded the introduction part to include these important comparisons (page 2, paragraph 2).

We also noticed the recently published work^{3,4} using skull optical clearing agents to improve the imaging performance by reducing skull scattering. This technique is mainly combined with conventional multi-photon microscopy to boost imaging resolution and penetration depth, albeit with restricted FOVs and lack of capability to measure flow velocity and RBC flux in a full-field manner. Furthermore, the skull optical clearing method could potentially be incorporated with RBC-aided WFLM to further improve imaging resolution and penetration depth. In the revised manuscript, we have included a discussion on skull optical clearing agents in the “Discussion” section (page 6, paragraph 3).

References:

1. Briers, J. D. & Webster, S. Laser speckle contrast analysis (LASCA): a non-scanning, full-field technique for monitoring capillary blood flow. *Journal of biomedical optics* **1**, 174-179 (1996).
2. Marchand, P. J. et al. Statistical parametric mapping of stimuli evoked changes in total blood flow velocity in the mouse cortex obtained with extended-focus optical coherence microscopy. *Biomedical optics express* **8**, 1-15 (2017).
3. Zhao, Y.-J. et al. Skull optical clearing window for in vivo imaging of the mouse cortex at synaptic resolution. *Light: Science & Applications* **7**, 17153-17153 (2018).
4. Li, D. et al. A Through-Intact-Skull (TIS) chronic window technique for cortical structure and function observation in mice. *eLight* **2**, 15 (2022).

Reviewers' Comments:

Reviewer #1:

Remarks to the Author:

The revised manuscript has adequately addressed my concerns regarding vessel diameters. The statistical analyses have been incorporated and are presented fairly. I am convinced that this novel tool has the power to analyze neuronal and vascular dynamics in model systems. I support its publication in Nature Communications.

Reviewer #2:

Remarks to the Author:

The authors have fully addressed my concerns. I think the paper is acceptable now.

Reviewer #3:

Remarks to the Author:

Thank the authors for their response. I have noted that the authors have included additional data and explanation to address the concerns I raised. I am happy with the part that the quality of the article has been well improved, and it is ready to be published.